# Cost-utility analysis of botulinum toxin type A versus oral drug treatment in patients with severe blepharospasm in Thailand

**Parima Hirunwiwatkul**[1,2], **Unchalee Permsuwan**[3,4*], **Sureerat Ngamkiatphaisan**[5], **Niphon Chirapapaisan**[6], **Jiruth Sriratanaban**[7]

1 Department of Ophthalmology, Faculty of Medicine, Chulalongkorn University, Bangkok, Thailand, 2 Department of Ophthalmology, King Chulalongkorn Memorial Hospital, Thai Red Cross Society, Bangkok, Thailand, 3 Department of Pharmaceutical Care, Faculty of Pharmacy, Chiang Mai University, Chiang Mai, Thailand, 4 Center for Medical and Health Technology Assessment (CM-HTA), Department of Pharmaceutical Care, Faculty of Pharmacy, Chiang Mai University, Chiang Mai, Thailand, 5 Research Center of Health Systems and Services, Faculty of Medicine, Chulalongkorn University, Bangkok, Thailand, 6 Department of Ophthalmology, Faculty of Medicine Siriraj Hospital, Mahidol University, Bangkok, Thailand, 7 Preventive and Social Medicine Department, Faculty of Medicine, Chulalongkorn University, Bangkok, Thailand

* unchalee.permsuwan@gmail.com

## Abstract

### Background

Blepharospasm is a chronic facial movement disorder affecting a person's ability to work, causing depression, pain, and a reduced quality of life (QoL). Botulinum toxin type A (BoNT-A) treatment can improve these conditions; however, its cost remains a significant barrier for inclusion of this indication into the Thai National List of Essential Medicine.

### Objectives

This study aimed to assess the cost-effectiveness of onabotulinumtoxinA (onaBoNT-A) and abobotulinumtoxinA (aboBoNT-A) treatment compared to oral medication treatment in patients with severe blepharospasm from a societal perspective.

### Methods

A cost-utility analysis using a two-part model was conducted to analyze lifetime costs and quality-adjusted life-years (QALYs). Inputs were mainly obtained from real-world evidence of 159 Thai patients with blepharospasm. Costs and outcomes were discounted at 3% annually and presented as 2023 value. Incremental cost-effectiveness ratios (ICERs) were estimated. Deterministic and probabilistic sensitivity analyses were also conducted.

### Results

In comparison to standard oral medication, both onaBoNT-A and aboBoNT-A incurred greater lifetime cost (3,055 USD, 2,889 USD vs 1,926 USD) while gaining additional QALYs (6.94 years, 6.94 years vs 6.53 years). The estimated ICERs were 2,722 USD/

**Data availability statement:** Data sharing for this study is restricted due to the involvement of 14 institutions. A de-identified dataset may be made available upon consensus from all participating centers. For data access requests, please contact the principal investigator, Dr.Parima, at hparima@gmail.com. Data can also be accessed through the Institutional Review Board (IRB) of the Faculty of Medicine, Chulalongkorn University, Bangkok, Thailand, at medchulairb@chula.ac.th, for researchers who meet the necessary criteria for access to confidential data.

**Funding:** This study was supported by the Food and Drug Administration, Ministry of Public Health, Thailand (Grant No.38/2567). This paper represents the views of the authors. This study was conducted at the request of the National List of Essential Medicine (NLEM). This manuscript is a part of the project "Economic evaluation and budget impact analysis of botulinum toxin type A versus oral drug treatment in patients with severe blepharospasm in Thailand" which was used to support the policy-making process under the Subcommittee for the Development of the NLEM in Thailand through the Health Economic Working Group (HEWG) but the HEWG is not responsible for the study findings and the dissemination of the findings.

**Competing interests:** The authors have declared that no competing interests exist.

QALY for onaBoNT-A and 2,323 USD/QALY for aboBoNT-A. Utility and cost of BoNT-A were important determinants in the sensitivity analysis.

## Conclusion

Among patients with severe blepharospasm, both onaBoNT-A and aboBoNT-A were considered a cost-effective strategy under the Thai willingness to pay threshold of 4,613 USD/QALY. Having aboBoNT-A was slightly more favorable due to lower cost, using a conversion ratio of 1U of onaBoNT-A: 3U of aboBoNT-A.

## 1. Introduction

Blepharospasm is a type of facial dystonia that presents significant challenges for patients due to manifesting involuntary, unpredictable muscle contractions around both eyes [1]. These bilateral spasms can escalate from frequent blinking to sustained eye closure, disrupting daily life and increasing the risk of accidents, particularly while driving [2]. In severe cases, the condition may lead to functional blindness, compounding safety concerns during routine activities. Many individuals report diminished confidence in essential areas such as work, driving, and social engagement due to these unpredictable episodes [3,4]. Beyond the physical symptoms, blepharospasm has been associated with psychological effects such as anxiety and depression, further reducing the overall quality of life for those affected [5–8].

Botulinum toxin type A (BoNT-A) remains the most effective treatment for blepharospasm, with established efficacy in reducing symptom severity and improving patients' quality of life with only minimal and temporary side effects [8–12]. In severe blepharospasm, oral medications, such as clonazepam, trihexyphenidyl, nortriptyline and baclofen, are generally ineffective, with response rates ranging from 9–25% [13], These medications are also often impractical due to the significant side effects such as sleepiness and drowsiness [14]. These side effects are particularly problematic for elderly patients and frequently lead to treatment discontinuation. Presently, access to BoNT-A treatment is not universal. While government and insurance coverage for BoNT-A is available in many countries. Thailand currently includes the treatment only for cervical dystonia, hemifacial spasm, and spasmodic dysphonia in the National List of Essential Medicines (NLEM) [15]. The omission of blepharospasm from this list is likely driven by concerns over high acquisition costs and perceived aesthetic uses.

Despite the evident potential to expand BoNT-A's indication for blepharospasm, an economic justification is required to achieve this in Thailand. A previous economic analysis conducted in Thailand in 2012 found that BoNT-A treatment for blepharospasm was not cost-effective [16]. However, this conclusion was based on a small set of international data, which may not fully reflect the local context in Thailand, particularly in terms of quality of life (QoL) improvements and costs specific to Thai patients. Recognizing the need for more accurate, locally relevant data, the Thai Neuro-Opthalmology Society undertook a prospective multicenter observational study from August 2020 to June 2021, enrolling daily-life affected (severe) blepharospasm patients from across 14 centers in Thailand to assess treatment efficacy, quality of life, and cost implications (data in S1Table) [8]. In response to these findings, the Health Economic Working Group (HEWG) under the sub-committee for NLEM development, proposed a cost-utility analysis comparing BoNT-A treatment with oral medication treatment for severe blepharospasm. This societal perspective analysis, grounded in local data, will provide critical evidence to guide policymakers in deciding whether to expand BoNT-A indications for blepharospasm. Such a decision would not only align Thailand's healthcare

system with international standards but also significantly improve patient outcomes by ensuring wider access to an effective treatment.

## 2. Methods

### 2.1. Cohort population

The cohort population in this study consisted of patients with blepharospasm. Eligible patients had a total Jankovic rating scale (JRS) of 6 or higher, which indicate severe symptoms that affect daily living. Subjects in this study were at least 61 years old.

### 2.2. Interventions and comparator

The study intervention was onabotulinumtoxinA (onaBoNT-A: Botox®) 30 units and abobotulinumtoxinA (aboBoNT-A: Dysport®) 90 units for a duration of approximately 12–16 weeks, depending on the patient's response. Due to differences in formulation and potency, the units of onaBoNT-A and aboBoNT-A are not directly comparable, with an approximate conversion ratio of 1 unit of onaBoNT-A to 3 units of aboBoNT-A for blepharospasm treatment. BoNT-A is typically administered once every 12–16 weeks, with dosages adjusted based on individual patient responses. Each session involves intramuscular injections at 4–10 sites around both eyes, depending on condition severity and physician preference.

The Thai Health Technology Assessment (HTA) guidelines recommend that the appropriate comparator should be the current practice, which are oral medications including clonazepam 75%, trihexyphenidyl 23% and others such as nortriptyline, baclofen 2%. The medication data was sourced from an unpublished survey from 35 Thai neuro-ophthalmologists and neurologists serving these patients.

### 2.3. Model structure

A two-part model was developed based on clinical evidence from our previous study [8]. The first component is a decision tree model (Fig 1A), followed by two lifetime Markov models (Fig 1B and 1C). The model ran on a 1-month cycle with a lifetime horizon. Starting from the decision tree model, the cohort population received either BoNT-A treatment or oral medication treatment. After treatment, survival was dependent on the transitional probability of treatment. Those who survived were recorded as responding or not responding to the BoNT-A treatment. The response group was classified into either the "good postpone" (GP) group, who required the BoNT-A treatment every 16 weeks, or the "good" (G) group, who required the BoNT-A treatment every 12 weeks. The non-responding group was classified into either the "poor" (P) group, who required the BoNT-A treatment every 12 weeks, or the "poor-postpone/quit" (PP) group, who would switch to the oral medications (comparator) in the next cycle.

The Markov model was composed of 3 health states: 1) JRS<6, 2) JRS≥6, and 3) death. Responding patients, both the GP and the P groups, would enter the Markov model at JRS<6. The P group would enter the JRS≥6 health state. The cohort would move between health states or remain in the same health state, relying on the transitional probabilities. Finally, all cohorts moved to death, which was the absorbing health state.

We assumed that the symptoms of patients who did not receive BoNT-A treatment remained severe with JRS≥6. Hence, the Markov model was comprised of 2 health states: 1) JRS≥6 and 2) death. The PP group and oral medication treatment group entered the Markov model at JRS≥6. The cohort remained in the same health state or moved to death in the next cycle. Finally, all cohorts moved to death.

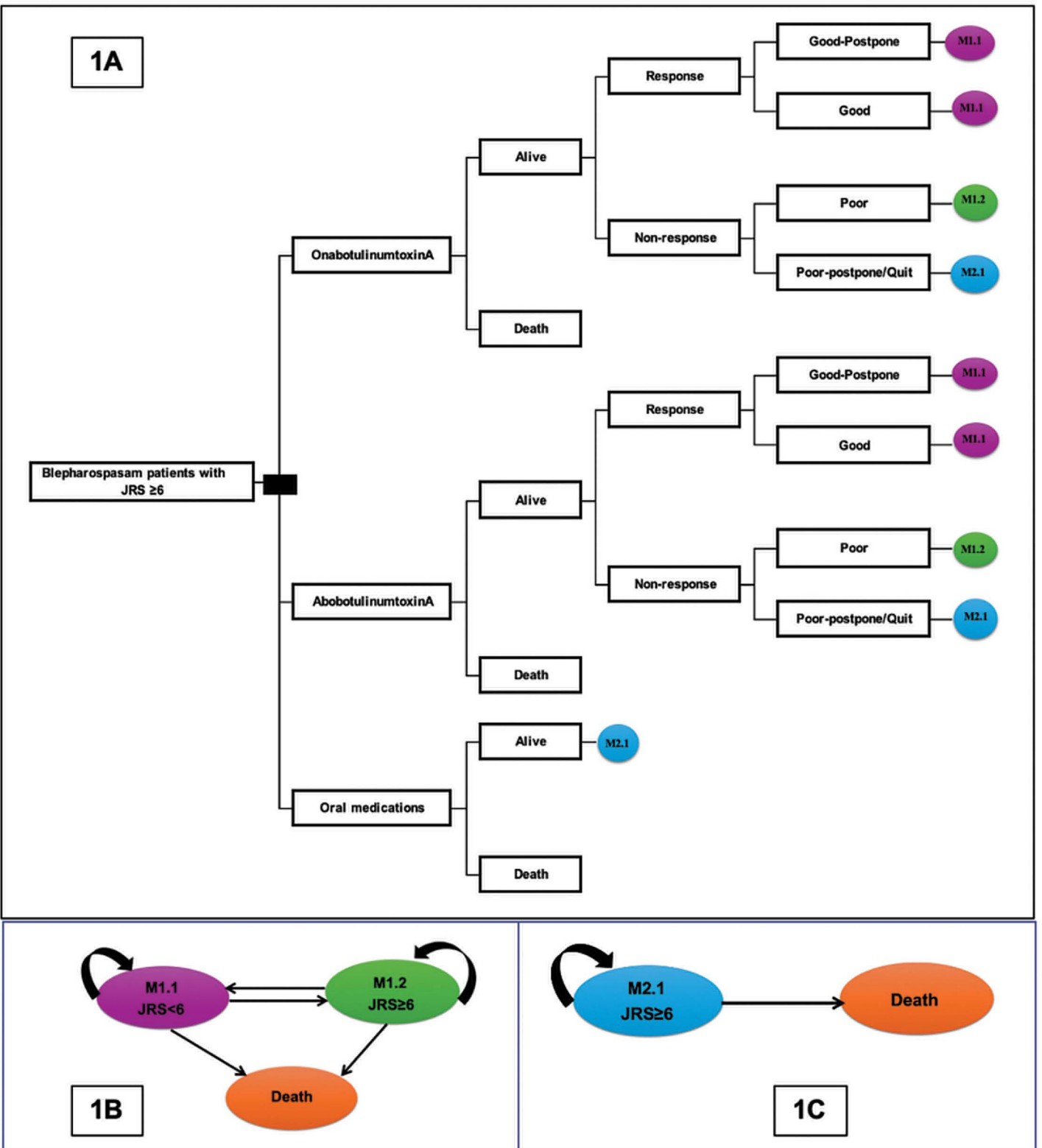

**Fig 1. Decision analytical model.**

## 2.4. Input parameters

Input parameters were obtained from a real-world study of Thai blepharospasm patients collected by the Thai Neuro-Ophthalmology Society [8]. In brief, this multicenter study enrolled 184 patients with daily-life-affecting blepharospasms from 16 clinics, either neuro-ophthalmology and neurology, and from 14 institutes, and the data was collected at baseline, at 1 month after initiating treatment, and 3 months after initiating treatment. QoL, severity grading, complications, accidents associated with the disease, and costs were collected before and after the BoNT-A treatment. All input details are shown in Table 1.

**2.4.1. Transitional probability.** Of the 184 patients, 159 (86%) had complete data necessary to derive the transitional probabilities. One month after receiving the BoNT-A treatment, all patients remained alive. Eight out of 159 patients (5%) did not respond to the BoNT-A treatment. For those who responded, 30 patients (20%) were classified as GP who needed the BoNT-A treatment every 16 weeks instead of 12 weeks. Among the non-responding group, 3 patients (38%) were classified as PP who were no longer in the BoNT-A treatment and moved to oral medication treatment in the next cycle.

Patients entered the Markov model based on the last health state in the decision tree model. The transitional probability from JRS≥6 to JRS<6 indicated that patients had improved symptom management from the BoNT-A treatment. Of all 159 patients, 151 patients (95%), including 30 GP patients and 121 G patients, saw their symptoms improve with JRS<6 on one month after receiving treatment. We assumed equal probability of response lasting for 3 months and 2 months for the GP group and the G group, respectively. We expected a drop in probability in the last month before patients were reinjected with the BoNT-A treatment. After the BoNT-A treatment for 3 months, 61 out of 159 patients (38%) remained JRS<6.

The transitional probability from JRS<6 to JRS≥6 represented patients experiencing disease worsening. Of all 30 patients in the GP group after receiving the BoNT-A treatment at the first month, 9 patients (30%) had JRS≥6 at month 3. We converted this 2-month probability to a 1-month rate using the formula $r = (\ln(1-p))/t$, where p denotes the probability, t is the time, and r represents rate. This rate was then transformed into a 1-month risk, which was equal to 16%, using the formula $p = 1-\exp(-rt)$, where p denotes the probability, r represents the rate and t is the time. We assumed that the monthly probability of worsening in the GP group remained constant from month 1 to month 3. All patients in the GP group received the BoNT-A treatment every 4 months. Since we lacked data for month 4 for the GP group, we assumed that the remaining 21 patients (70%) had JRS≥6 at month 4. For the G group, the transitional probabilities from JRS<6 to JRS≥6 were derived in the same manner as the GP group. Risk of RS<6 to JRS≥6 for month 1 and month 2 equalled 44%, and month 3 equalled 69%.

In this study, we included the costs of motor vehicle accidents for blepharospasm patients such as C-spine fracture, limb fracture, blunt abdominal injury, and non-incapacitating injury treatment. Risks of such accidents were obtained from a previous published study as shown in Table 1 and Table 2 [2].

**2.4.2. Costs.** Given that the present study adopted a societal perspective, we included both direct medical costs and direct non-medical costs. Indirect costs were excluded to avoid double counting of both the cost and the effect of the intervention for a cost-utility analysis based on the Health Technology Assessment Guidelines in Thailand [18]. Direct medical costs included drugs, outpatient visits, and treatment costs associated with injuries related to accidents due to vision impairment.

The costs of onaBoNT-A and aboBoNT-A were sourced from auction prices from the Drug and Medical Supply Information Center, Ministry of Public Health. The cost of outpatient department (OPD) services and injection fees were obtained from the average prices surveyed from the multicenter study hospitals which were in accordance with the service rate set by

**Table 1. Input parameters.**

| Parameters | Mean | SE | Distribution | References |
|---|---|---|---|---|
| Transitional probabilities: Decision tree model | | | | |
| *BoNT-A treatment* | | | | Primary data collection [8] |
| - Probability of alive | 1.00 | NA | Fixed | |
| - Probability of nonresponse | 0.05 | 0.02 | Beta | |
| - Probability of GP | 0.20 | 0.03 | Beta | |
| - Probability of PP | 0.38 | 0.16 | Beta | |
| *Oral medication* | | | | |
| - Probability of alive | 1.00 | NA | Fixed | |
| Transitional probabilities: Markov model | | | | |
| Probability of JRS≥6 to JRS<6 | 0.95 | 0.02 | Beta | |
| - Month 1-3 for GP group | 0.38 | 0.04 | Beta | |
| - Month 4 for GP group | | | | |
| Probability of JRS≥6 to JRS<6 | 0.95 | 0.02 | Beta | |
| - Month 1-2 for G group | 0.38 | 0.04 | Beta | |
| - Month 3 for G group | | | | |
| Probability of JRS<6 to JRS≥6 | 0.16 | 0.01 | Beta | |
| - Month 1-3 for GP group | 0.70 | 0.08 | Beta | |
| - Month 4 for GP group | | | | |
| Probability of JRS<6 to JRS≥6 | 0.44 | 0.02 | Beta | |
| - Month 1-2 for G group | 0.69 | 0.04 | Beta | |
| - Month 3 for G group | | | | |
| Probability of accidence (per 1 million population) | | | | |
| *BoNT-A treatment* | | | | Hwang WJ [2] |
| - C-spine fracture treatment | 1.00 | 0.05 | Beta | |
| - Limb fracture treatment | 293.86 | 14.99 | Beta | |
| - Blunt abdominal injury | 1.00 | 0.05 | Beta | |
| - Non-incapacitating injury treatment | 898.31 | 45.83 | Beta | |
| *Oral medication* | | | | |
| - C-spine fracture treatment | 353.35 | 18.03 | Beta | |
| - Limb fracture treatment | 1446.18 | 73.78 | Beta | |
| - Blunt abdominal injury | 353.35 | 18.03 | Beta | |
| - Non-incapacitating injury treatment | 4211.51 | 214.87 | Beta | |
| Costs (USD) | | | | |
| *Direct medical costs* | | | | |
| - AbobotulinumtoxinA (Dysport®) (per injection) | 58.17 | 5.53 | Gamma | Auction prices in the Drug and Medical Supply Information Center, Ministry of Public Health |
| - OnabotulinumtoxinA (Botox®) (per injection) | 63.25 | 6.46 | Gamma | |
| - Outpatient visit for BoNT-A treatment (per month) | 2.02 | 0.21 | Gamma | Primary data collection from Thai neuro-ophthalmology society |
| - Oral medications (per month) | 1.73 | 0.18 | Gamma | |
| - Outpatient visit for oral medication treatment (per month) | 0.48 | 0.05 | Gamma | |
| - C-spine fracture treatment | 2165.44 | 220.96 | Gamma | The average medical expenses for complications from the joint diagnosis group at King Chulalongkorn Memorial Hospital |
| - Limb fracture treatment | 6919.65 | 706.09 | Gamma | |
| - Blunt abdominal injury | 1207.58 | 123.22 | Gamma | |
| - Non-incapacitating injury treatment | 4.04 | 0.41 | Gamma | |

*(Continued)*

**Table 1.** (Continued)

| Parameters | Mean | SE | Distribution | References |
|---|---|---|---|---|
| *Direct non-medical costs* | | | | Primary data collection and standard cost list [17] |
| - Food and travel of GP group | 6.98 | 0.71 | Gamma | |
| - Food and travel of G group | 6.28 | 0.64 | Gamma | |
| - Food and travel of P group | 6.93 | 0.71 | Gamma | |
| - Food and travel of PP group | 4.96 | 0.51 | Gamma | |

Abbreviation: GP, good postpone; G, good; P, poor; PP, poor postpone; SE, standard error.

**Table 2. Utility data.**

| | Markov model (3 health states) | | | | Markov model (2 health states) |
|---|---|---|---|---|---|
| | 16-week BoNT treatment | | 12-week BoNT treatment | | Oral medication treatment |
| | JRS<6 | JRS≥6 | JRS<6 | JRS≥6 | JRS≥6 |
| Month 1 | 0.86 | 0.84 | 0.82 | 0.81 | 0.75 |
| Month 2 | 0.86 | 0.81 | 0.82 | 0.81 | |
| Month 3 | 0.87 | 0.71 | 0.79 | 0.71 | |
| Month 4 | 0.85 | 0.63 | | | |

theThailand Comptroller General's department. The cost of accidents was derived from the average medical expenses for complications from the joint diagnosis group at King Chulalongkorn Memorial Hospital. For other complications in both groups that were mild and recovery did not involve treatment, patients did not incur additional costs.

Direct non-medical costs, such as cost of transportation to the institutes and related food, were directly collected from the 159 patients. The traveling costs did not significantly differ among the GP, G, and P groups, while the PP group had the lowest traveling costs. Cost of food was obtained from the standard cost list in Thailand [17]. Direct non-medical costs of GP, G, P, and PP groups totalled 6.98 USD, 6.28 USD, 6.93 USD, and 4.96 USD, respectively (Table 1).

All cost data were adjusted for inflation based on the medical care section of Thailand's consumer price index in 2023 [19]. The costs were converted into USD at a rate of 34.68 THB per USD, as of 22 December 2023 [20].

**2.4.3. Utility.** The utility data were adopted from the previous Thai real-world study [8], which collected data from the 159 severe blepharospasm patients using the Thai version of EQ-5D-5L. All utility values are shown in Table 2.

## 2.5. Study outcomes

Study outcomes were total lifetime cost, life-year (LYs), quality-adjusted life-years (QALYs) (product of utility and LYs), incremental costs, LY gained, QALYs gained, and incremental cost-effectiveness ratio (ICER).

## 2.6. Data analyses

**2.6.1. Base-case analysis.** In the base-case analysis, the ICER was calculated as the difference in lifetime costs between BoNT-A treatment and oral medication treatment divided by the difference in their outcomes. The costs and outcomes were considered over a lifetime horizon and were discounted at an annual rate of 3% according to the Thai HTA guidelines [21]. The local

threshold of 4,613 USD/QALY or 160,000 THB/QALY, which is about 1.2 times per capita gross national income [22], was used to justify the cost-effectiveness of the new intervention.

**2.6.2. Sensitivity analyses.** A variety of sensitivity analyses were conducted to determine whether the results were robust to the differences that arise from uncertainty in the parameters. In the one-way sensitivity analysis, each parameter was individually varied by its specified range. In cases where specific ranges were unavailable, a variation of 10% from mean was used for transitional probabilities and utilities, and a variation of 20% from mean was used for costs. Furthermore, the discount rates for costs and outcomes were varied from 0% to 6% based on the recommendation of the Thai HTA guidelines [21]. The results of the one-way sensitivity analyses were displayed as a tornado diagram.

In addition, a probabilistic sensitivity analysis (PSA) was conducted whereby individual sets of parameter values were drawn from appropriate statistical distributions (Table 1), with results generated for 1000 simulation runs. The PSA results were displayed as a scatterplot on a cost-effectiveness plane and a cost-effectiveness acceptability curve (CEAC). The CEAC represented the probability of BoNT-A treatment being cost-effective, compared with oral medication treatment, for different defined willingness-to-pay thresholds.

## 2.7. Ethics Approval

The Institutional Review Board of the Faculty of Medicine, Chulalongkorn University, Bangkok, Thailand, has exempted this study from compliance with the international guidelines for human research protection as Declaration of Helsinki, The Belmont Report, CIOMS Guidelines, International Conference on Harmonization in Good Clinical Practice (ICH-GCP) and 45CFR 46.101(b). We retrieved the data on May 8, 2023. The COE No. is 030/2023. The authors cannot identify individual participants because we used deidentified data from pooled multi-center data.

**Consent of Participant** Not applicable.
**Consent for Publication (from Patients/Participants)** Not applicable.
**Code Availability** Not applicable.

## 3. Results

### 3.1. Base-case results

Our model predicted that patients with severe blepharospasm treated with BoNT-A treatment, both onaBoNT-A and aboBoNT-A, incurred greater total lifetime costs, but gained more QALYs than those who did not receive the BoNT-A treatment. Data are presented in Table 3: QALY was 6.94 in the BoNT-A treatment group and the 6.54 in oral medication group. OnaBoNT-A treatment had higher total lifetime costs compared to aboBoNT-A treatment (3,055 USD vs 2,889 USD). Compared to oral medication treatment, incremental costs of onaBoNT-A treatment and aboBoNT-A treatment were 1,129 USD, and 963 USD, respectively, while both onaBoNT-A treatment and aboBoNT-A treatment also had an equal incremental QALY of 0.41 years. As a result, when compared with oral medication treatment, onaBoNT-A treatment had higher ICER than aboBoNT-A treatment (2,722 USD/QALY vs 2,323 USD/QALY). The details are shown in Table 2.

### 3.2. Sensitivity analysis results

The scatter plot on the cost-effectiveness plane demonstrated that all iterations fell on the upper quadrants meaning that treatment with BoNT-A had higher costs than oral medication. The ICER of base-case results represented as the red dot on Figs 2A and 3A fell within the acceptable threshold in Thailand.

**Table 3. Base-case results.**

|  | Oral medication treatment | OnaBoNT-A (Botox') | AboBoNT-A (Dysport') |
|---|---|---|---|
| Lifetime total cost (USD) (%) | 1,926 (100%) | 3,055 (100%) | 2,889 (100%) |
| - Drugs | 181 (9.41%) | 2,064 (67.58%) | 1,899 (65.73%) |
| - Out-patient visit | 50 (2.61%) | 198 (6.49%) | 198 (6.86%) |
| - Accident treatment | 1,175 (61.02%) | 100 (3.29%) | 100 (3.47%) |
| Total life-year | 8.73 | 8.74 | 8.74 |
| Total QALY | 6.53 | 6.94 | 6.94 |
| Incremental cost (USD) | Reference | 1,129 | 963 |
| Incremental QALY | Reference | 0.41 | 0.41 |
| ICER (USD/QALY) |  | 2,722 | 2,323 |

Abbreviation: ICER, incremental cost-effectiveness ratio; QALY, quality adjusted life-years.

The cost-effectiveness acceptability curve after 1,000 simulations showed the relationship between probability of each treatment being cost-effective versus the various levels of willingness to pay per one additional QALY. With the Thai ceiling threshold of 4,613 USD/QALY, the probability that onaBoNT-A treatment and aboBoNT-A treatment was cost-effective for the treatment of patients with severe blepharospasm was 73% (Fig 2B) and 77%, (Fig 3B) respectively.

The results of the one-way sensitivity analysis using a tornado diagram revealed that the utility of JRS≥6 of poor state at month 3 had the strongest impact on the ICER estimate. This was followed by the utility of JRS≥6 of poor-postpone state and the cost of BoNT-A treatment as shown in Fig 4.

## 4. Discussion

Blepharospasm is not merely a cosmetic concern, since it significantly impairs quality of life, reduces abilities, and can lead to accidents [2,5–8]. The use of botulinum toxin is primarily aimed at reducing the risk of accidents and eliminating the costs associated with treating such accidents. By decreasing spasms around the eyes, botulinum toxin allows patients to open their eyes more easily, leading to increased confidence in activities such as going outside, driving, working, and engaging in social activities. This improvement in quality of life alleviates concerns about sudden closure of both eyes, which could cause accidents both outdoors and indoors. Due to the high acquisition cost of the BoNT-A treatment in Thailand, economic evidence is needed to make a policy recommendation to support the widespread adoption of the BoNT-A treatment in the whole country. The findings of this study, which completed a cost-utility analysis of the BoNT-A treatment (onaBoNT-A and aboBoNT-A) versus oral drug treatment in patients with severe blepharospasm, showed the BoNT-A treatment to be a cost-effective treatment for patients with severe blepharospasm with an ICER of 2,722 USD/QALY gained (94,409 THB/QALY gained) for onaBoNT-A, and 2,323 USD/QALY gained (80,568 THB/QALY gained) for aboBoNT-A. A chance of being a cost-effective strategy at a willingness to pay in Thailand was slightly higher for aboBoNT-A compared to onaBoNT-A (77% vs 73%).

The findings of this study were in line with the results of a UK study [23], which assessed the cost-effectiveness of BoNT-A versus placebo treatment. The results of the UK study demonstrated that BoNT-A treatment incurred higher total cost and gained higher QALY than placebo. The estimated ICER was below the acceptable threshold of the country. The main cost difference between the treatment group and placebo was that of the BoNT-A injection itself.

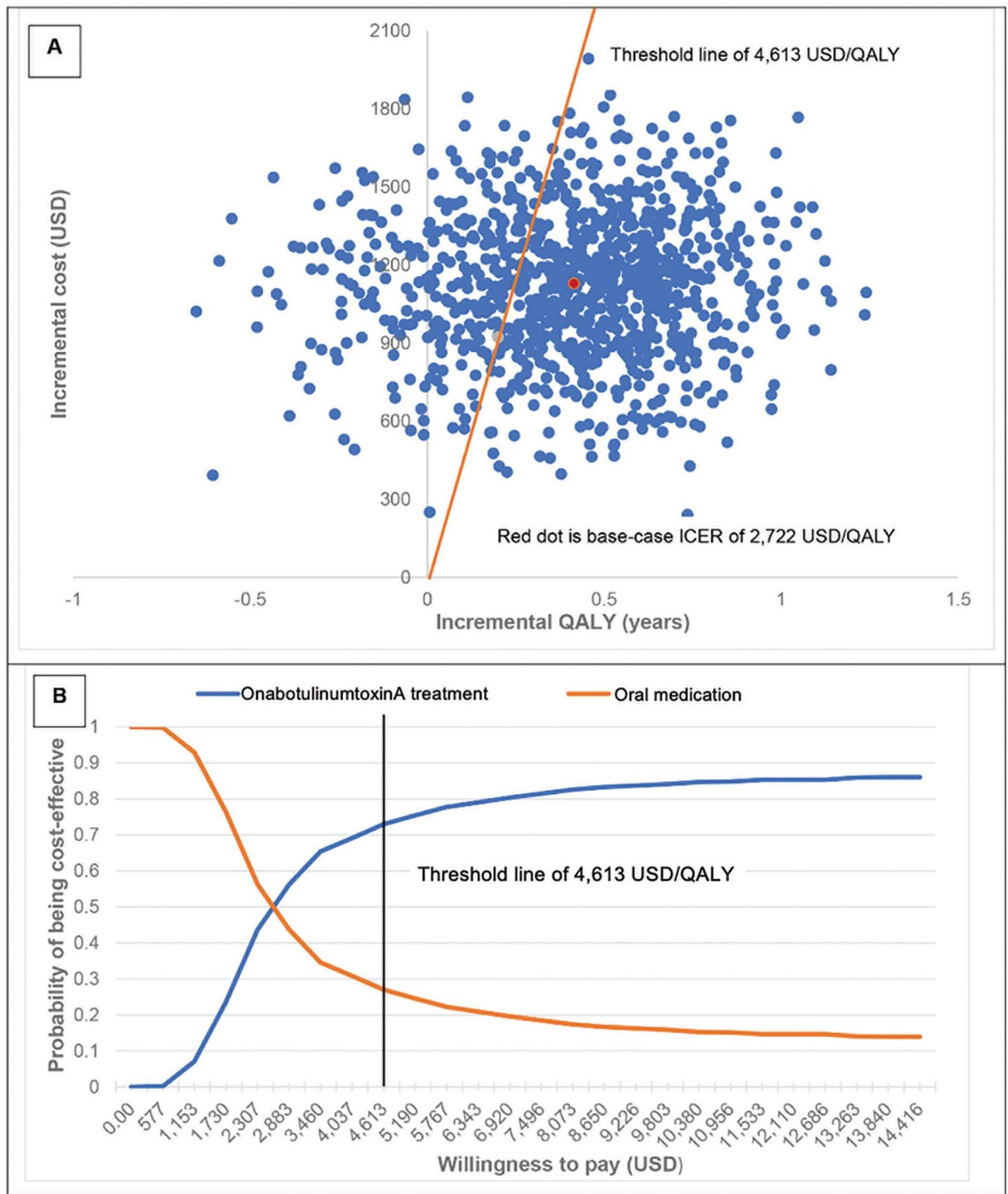

**Fig 2. Probabilistic sensitivity analysis of onabotulinumtoxinA treatment vs oral medication treatment (A: scatter plot on cost-effectiveness plane, B: cost-effectiveness acceptability curve).**

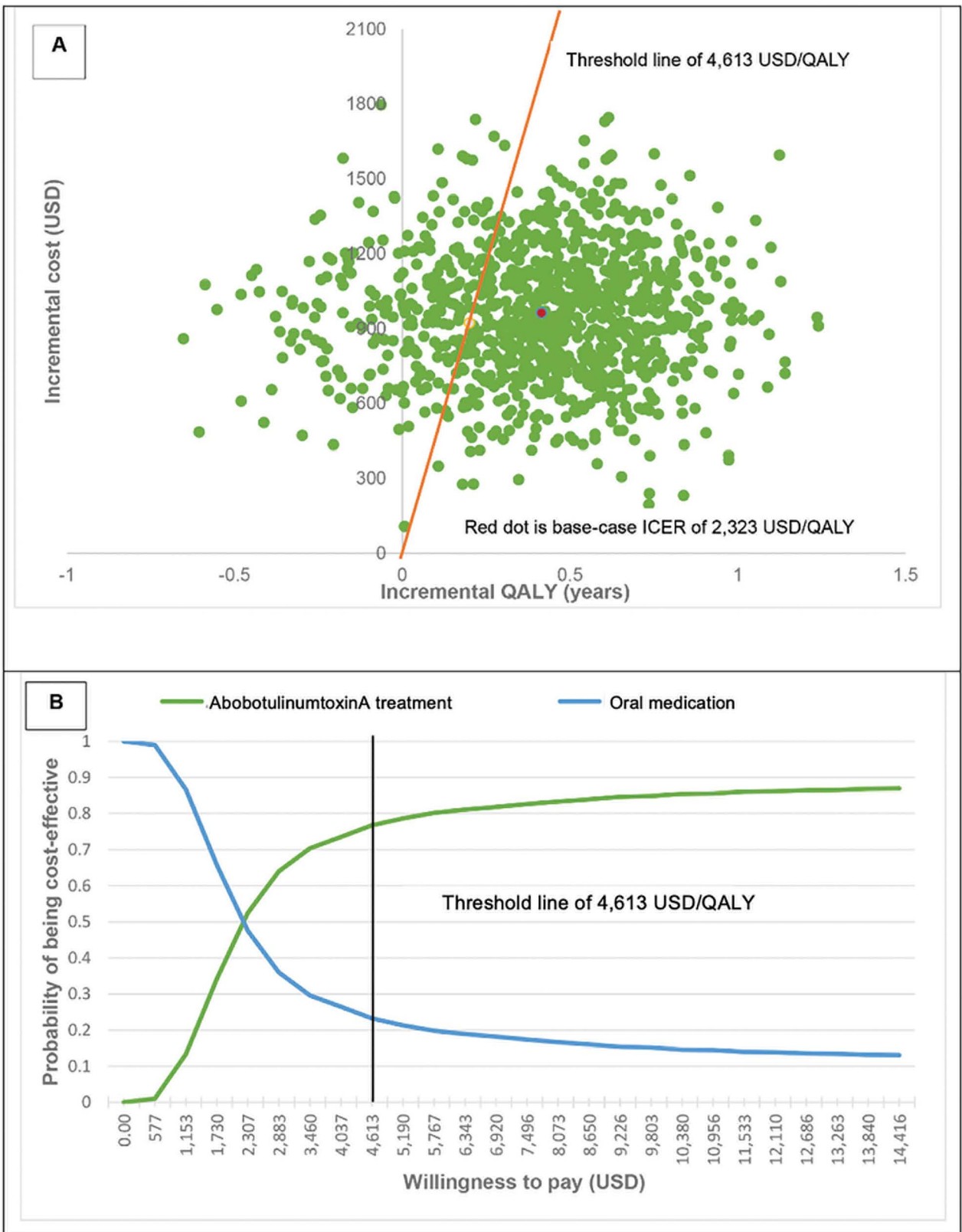

**Fig 3. Probabilistic sensitivity analysis of abobotulinumtoxinA treatment vs oral medication treatment (A: scatter plot on cost-effectiveness plane, B: cost-effectiveness acceptability curve).**

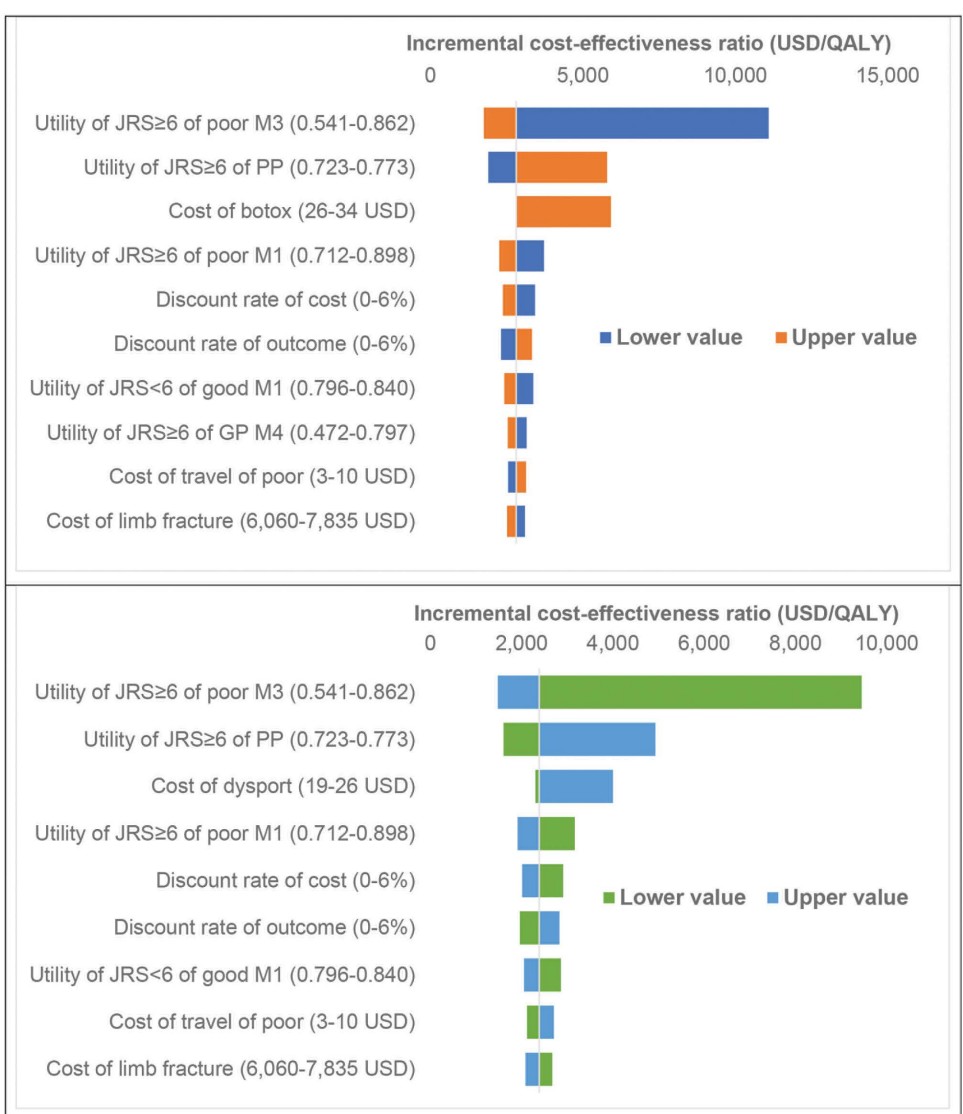

**Fig4. Tornado diagram (A: OnaBoNT-A treatment vs oral medication treatment, B: AboBoNT-A treatment vs oral medication treatment).**

The findings of the present analysis of the BoNT-A treatment are, however, not consistent with the result of an older 2012 cost-utility study of the BoNT-A treatment in Thailand [16] which found the BoNT-A treatment not to be cost-effective. Differences might have arisen due to several factors. Firstly, the cost of accident/injury treatment was not included in the previous study while it was included in our current study. Without the BoNT-A treatment, cost of accident treatment contributed around 61% of total lifetime total cost. Given the BoNT-A treatment, the contribution of accident costs greatly declined to about 3% (Table 3). Secondly, the acquisition costs of both onaBoNT-A and aboBoNT-A in the present study were 20%-30% lower than those in the previous study. Thirdly, the models used were different between the two studies. Finally, input parameters of the previous economic study, such as transitional probabilities and utility, were obtained from publications from other countries, while in this study inputs were based on real-world data from Thai blepharospasm patients.

Some strengths and limitations were present in our study. Firstly, the developed decision model used in this study captured long-term costs, life-years, and QALYs. Almost all inputs were obtained from the real-world 159 patients with severe blepharospasm across 16 centers in Thailand. However, the observational study collected the data for only 3 months. We used this data to derive the monthly transitional probability by assuming the same pattern of a 12-week BoNT-A treatment. Secondly, this study incorporated the cost of accident treatment into the analysis which is a key strength of our study since we looked at total societal costs, not just cost of the procedure. Our study better captures the real benefits from the BoNT-A treatment for patients with severe blepharospasm, in which accidents occur quite frequently. Further study should recruit a larger sample size or collect longer term QoL data.

## 5. Conclusions

Our findings suggest that, both onaBoNT-A and aboBoNT-A are cost-effective for patients with severe blepharospasm at the local threshold of 4,613 USD/QALY. Compared with oral medication treatment, aboBoNT-A treatment showed a lower ICER than onaBoNT-A treatment (2,323 USD/QALY vs 2,722 USD/QALY), with a conversion ratio of 1:3 for onaBoNT-A:aboBoNT-A, due to lower acquisition drug costs. These results support the recommendation of adding the indication of severe blepharospasm for BoNT-A treatment in the Thai NLEM.

## Supporting information

**S1 Table. Demographic data of patient references.**
(DOCX)

## Acknowledgements

The authors thank the Royal College of Thai Ophthalmologist Thai Neuro-Ophthalmology Society and co-authors of health-related quality of life of daily-life-affected benign essential blepharospasm: multi-center observational study: Wajamon Supawatjariyakul, Supharat Jariyakosol, Supanut Apinyawasisuk, Yuda Chongpison, Priya Jagota, Nipat Aui-aree, Juthamat Witthayaweerasak, Suwanna Setthawatcharawanich, Kitthisak Kitthaweesin, Piyawadee Chaimongkoltrakul, Poramaet Laowanapiban, Linda Hansapinyo, Suthida Panpitpat, Sireedhorn Kurathong, Jirat Nimworaphan, Suntaree Thitiwichienlert, Kavin Vanikieti, Narong Samipak, Worapot Srimanan, Nattapong Mekhasingharak, and Pareena Chaitanuwong, for set up and collecting treatment response data, quality of life measures, and costs of BoNT-A treatment of Thai daily-life affected blepharospasm patients. These are essential data for economic analysis in this article.

## Author contributions

**Conceptualization:** Parima Hirunwiwatkul, Unchalee Permsuwan, Sureerat Ngamkiatphaisan, Niphon Chirapapaisan, Jiruth Sriratanaban.

**Data curation:** Parima Hirunwiwatkul, Unchalee Permsuwan.

**Formal analysis:** Unchalee Permsuwan.

**Funding acquisition:** Parima Hirunwiwatkul.

**Investigation:** Parima Hirunwiwatkul, Unchalee Permsuwan, Sureerat Ngamkiatphaisan.

**Methodology:** Parima Hirunwiwatkul, Unchalee Permsuwan, Sureerat Ngamkiatphaisan, Niphon Chirapapaisan, Jiruth Sriratanaban.

**Project administration:** Parima Hirunwiwatkul, Unchalee Permsuwan.

**Supervision:** Jiruth Sriratanaban.

**Validation:** Parima Hirunwiwatkul, Unchalee Permsuwan, Sureerat Ngamkiatphaisan, Niphon Chirapapaisan, Jiruth Sriratanaban.

**Writing – original draft:** Parima Hirunwiwatkul, Unchalee Permsuwan.

**Writing – review & editing:** Parima Hirunwiwatkul, Unchalee Permsuwan, Sureerat Ngamkiatphaisan, Niphon Chirapapaisan, Jiruth Sriratanaban.

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
