## [Decision Letter · Decision Letter 0]

4 Oct 2024

PONE-D-24-22961Cost-utility analysis of botulinum toxin type A versus oral drug treatment in patients with severe blepharospasm in ThailandPLOS ONE

Dear Dr. permsuwan,

Thank you for submitting your manuscript to PLOS ONE. After careful consideration, we feel that it has merit but does not fully meet PLOS ONE’s publication criteria as it currently stands. Therefore, we invite you to submit a revised version of the manuscript that addresses the points raised during the review process.

**ACADEMIC EDITOR: **

Pls make point-to-point response to the reviewers' comments.

We look forward to receiving your revised manuscript.

Kind regards,

Gang Qin, PhD, MD

Academic Editor

PLOS ONE

Reviewers' comments:

Reviewer's Responses to Questions

**Comments to the Author**

1. Is the manuscript technically sound, and do the data support the conclusions?

Reviewer #1: Yes

Reviewer #2: Yes

Reviewer #3: Partly

2. Has the statistical analysis been performed appropriately and rigorously?

Reviewer #1: Yes

Reviewer #2: Yes

Reviewer #3: Yes

3. Have the authors made all data underlying the findings in their manuscript fully available?

Reviewer #1: Yes

Reviewer #2: No

Reviewer #3: No

4. Is the manuscript presented in an intelligible fashion and written in standard English?

Reviewer #1: Yes

Reviewer #2: Yes

Reviewer #3: Yes

5. Review Comments to the Author

Reviewer #1: 1. Does the analysis compare the outcomes of patients treated with Botulinum toxin A (BoNT-A) to those receiving either oral medications or no BoNT-A treatment, and are "no BoNT-A" and "oral medications" considered the same and used interchangeably? Please clarify and be consistent throughout the manuscript

2. Are the health effects measured in QALYs for both onaBoNT-A and aboBoNT-A? Please provide a table that separates the data for each agent.

3. The introduction does not specify the perspectives being considered. Please clarify and include this information

4. The introduction mentions that oral medications were not effective or practical, but it lacks detail and supporting data. Please provide further elaboration, including specific data or evidence that demonstrates the limitations of oral medications.

5. Please elaborate on the prospective multicenter observational studies, specifying whether they were designed with two arms or three arms. Additionally, clarify when and where data on oral medication were included in the study.

6. Please include the cost of oral medications in Table 1 to facilitate understanding, and ensure the source of this cost information is clearly indicated within the table.

7. Please provide a citation for data on accidents resulting from severe blepharospasm

8. Please add the p-value to Table 3 for easier interpretation

9. Was the abbreviation "JRS" mentioned in full before it was used in the manuscript?

10. Please cite for the definition of blepharospasm.

11. Please suggest ways to improve the sensitivity for utility of JRS in future analyses etc.

Reviewer #2: Thank you for the opportunity to review this article. The study is well-designed and written. However, some comments should be considered by the authors.

1. The authors should add more information in the introduction about the national burden of Blepharospasm in Thailand. This will justify the study question

2. In the abstract, what does the author mean by hybrid model? This is not well-known in pharmacoeconomic studies. If they mean decision tree followed by Markov then please mention it.

3. Is there any population receive supportive care? It is important to mention what is the standard practice in Thailand.

4. How you confirmed that the study sample is generalizable to Thailand population? Please add a table for patient demographics.

Reviewer #3: This was an interesting study. One of the issues I found was that the two treatment regimens for BoNT-A treatment were not separate nodes in the decision tree model but were instead combined. Since they are different alternatives of the BoNT-A treatment, each arm can be represented by their own node, with relevant probabilities calculated from the study data. Secondly, the decision tree for the comparator treatment assumed that there was no therapeutic effect of the comparator treatment. How is this different than assuming that the patient took no other treatment? Finally, the BoNT-A treatments were not adequately described which caused further confusion when the authors talked about months of treatment, and subsequent follow-up treatment.

Page 5, Line 108 to 109:

Please define terms before using abbreviations. What does JRS stand for?

Secondly, the sentence needs to be rephrased for coherence. Do you mean, “Eligible patients had a total JRS score of 6 or higher, which indicate severe symptoms that affect daily living”.

Page 6, Line 112 to 114:

Could you please be more descriptive and specific about the interventions? The way this paragraph is written, it gives the impression that patients were given 30 units of onaBoNT-A per day for 12 weeks and up to a total of 16 weeks. It also gives the impression that patients were given 90 units of aboBoNT-A per day for 12 weeks and up to a total of 16 weeks. Is that the dosage of this intervention? What is the route of administration? Is the treatment actually just one injection that is administered once, and a subsequent injection is administered after 12 or 16 weeks based on the patient response.

Page 6, line 115 to 119:

Which comparator treatments are considered in this study, and how are they incorporated into the model in the subsequent section of the manuscript?

Figure 1A:

BoNT-A treatment arm: Why were both interventions (onaBoNT-A and aboBoNT-A) not included as separate arms in the decision tree mode? Is there no difference between the two?

No BoNT-A treatment arm: How does this arm take into consideration the appropriate comparator from current practice? How is it different from no treatment at all?

Page 7, line 143:

Is there no evidence for the efficacy of the comparator treatment? Is there a rationale to assume that patients that did not receive BoNT-A treatment remained at a JRS score of 6 or higher even with alternative comparator treatments? This assumption has implications for figures 1A, 1B and 1C.

Page 7, line 153:

Please explain clearly what it means to say, “collected data at month 0, 1 and 3 after treatment”. How is data collected at 0 months after treatment? Do you mean data was collected at baseline, at 1 month after initiating treatment, and 3 months after initiating treatment?

Page 10, paragraph 1 and 2:

What does it mean “After receiving the BoNT-A treatment for 1 month”? Does it mean that each patient received a daily dose of BoNT-A treatment for a whole month? That is what the sentence means, but I don’t think that is what is intended. Also, since there are two BoNT-A treatments (onaBoNT-A and aboBoNT-A), how does treatment differ between the two arms?

It is stated that of the 159 patients, 151 (95%) went from a JRS=>6 to a JRS<6 after the first month of treatment. This sentence means that after receiving treatment for one month, the JRS score went from a baseline of JRS=>6 before treatment initiation, to a score of JRS<6 at the end of 1 month of treatment. It is also stated that the GP group needs BoNT-A treatment every 16-weeks. If we assume that January is the one month of treatment, the GP group would need a second regimen of BoNT-A treatment on June 1st, when 16 weeks have elapsed since the first month of treatment. Is this correct? I don’t believe that is what the authors are trying to convey. If in reality, the case is that the GP group would need a second regimen of BoNT-A treatment on May 1st, then I believe the correct way to phrase your statement regarding treatment would be “One month after receiving the BoNT-A treatment”, instead of “after the first month of receiving treatment”. Clarification is needed in this regard, and if the authors adequately describe how the BoNT-A treatment is administered, the confusion will be cleared.

Page 12, line 207:

Do you mean “Comptroller General…”

Page 14, line 249:

Do you mean "with results generated…”

Page 15, line 160-161:

“COE No. is….”. “…..because we used deidentified data from the pooled multi-center data.”

Page 15, line 268-269:

“…both onaBoNT-A and aboBoNT-A, incurred greater total lifetime cost, but gained fewer-life years and QALYs than those who did not receive BoNT-A treatment”. How does your study show fewer QALYs were gained when your results show an incremental QALY (0.41) with BoNT-A treatment?

Page 16, line 281:

“…fell within the acceptable threshold in Thailand.”

6. PLOS authors have the option to publish the peer review history of their article (what does this mean? ). If published, this will include your full peer review and any attached files.

**Do you want your identity to be public for this peer review?** For information about this choice, including consent withdrawal, please see our Privacy Policy .

Reviewer #1: No

Reviewer #2: No

Reviewer #3: No

---

## [Author Response · Author response to Decision Letter 1]

12 Nov 2024

PONE-D-24-22961

Cost-utility analysis of botulinum toxin type A versus oral drug treatment in patients with severe blepharospasm in Thailand

Dear Editors and Reviewers,

Thank you for your time and kind comments. Here is our team’s explanations and additional contents which already added in the revised manuscript.

Editor comment:

Thank you for the opportunity to review this article. The study is well-designed and written. However, some comments should be considered by the authors.

>> Thank you for your helpful comment.

1. The authors should add more information in the introduction about the national burden of Blepharospasm in Thailand. This will justify the study question

Response: (Introduction part)

We revised introduction part to show the background of this study and research question clearer than the first draft, include national burden of blepharospasm in Thailand

2. In the abstract, what does the author mean by hybrid model? This is not well-known in pharmacoeconomic studies. If they mean decision tree followed by Markov then please mention it.

Response: (Abstract: Method part)

We have revised the word from a hybrid model to a two-part model, which is similar to the method section.

A two-part model in this study represents the decision tree and Markov model.

3. Is there any population receive supportive care? It is important to mention what is the standard practice in Thailand.

Response: (Introduction part)

This is model-base study by use cohort population in the model. Use cohort population from multicenter study: supportive care (oral medical treatment) and BoNT-A treatment.

4. How you confirmed that the study sample is generalizable to Thailand population? Please add a table for patient demographics.

Response: (Introduction part)

The data were recruit from the 14 nationwide medical centers across the country. These centers represent the main providers of botulinum toxin treatments for blepharospasm in Thailand. The demographic data of reference source was showed in S1table.

We provide the table of demographic data of the patients as S1Table: Demographic data of patient references.

Response: Done. Thank you.

Response: Done. Thank you.

Response: Done. Thank you.

Reviewers' comments:

>> Thank you for your valuable comments on the manuscript. We appreciate your insights and feedback, which have been instrumental in enhancing the quality of our work.

Reviewer #1: Thank you for your comments.

1. Does the analysis compare the outcomes of patients treated with Botulinum toxin A (BoNT-A) to those receiving either oral medications or no BoNT-A treatment, and are "no BoNT-A" and "oral medications" considered the same and used interchangeably? Please clarify and be consistent throughout the manuscript

Response:

We have changed the term “no BoNT-A treatment” to “oral medications” for the whole manuscript. The analysis compared the outcomes of patients treated with Botulinum toxin A (BoNT-A) to those receiving oral medications as control.

2. Are the health effects measured in QALYs for both onaBoNT-A and aboBoNT-A? Please provide a table that separates the data for each agent.

Response:

In analysis, we assume onaBoNT-A and aboBoNT-A are not different effects as in previous study.

Hirunwiwatkul P, Supawatjariyakul W, Jariyakosol S, Apinyawasisuk S, Sriratanaban J, Chongpison Y, et al. Health-related quality of life of daily-life-affected benign essential blepharospasm: Multi-center observational study. PLOS ONE. 2023;18(3):e0283111.

3. The introduction does not specify the perspectives being considered. Please clarify and include this information

Response: (Introduction part)

We use societal perspective according to HTA guideline requirements. There are in objective part of abstract. We also added it in the introduction part.

4. The introduction mentions that oral medications were not effective or practical, but it lacks detail and supporting data. Please provide further elaboration, including specific data or evidence that demonstrates the limitations of oral medications.

Response: (Introduction part)

In severe blepharospasm, oral medications, such as clonazepam, trihexyphenidyl, nortriptyline and baclofen, are generally ineffective, with response rates ranging from 9-25%[Ref], and often impractical due to the significant side effects, especially sleepiness and drowsiness[Ref]. These side effects are particularly problematic for elderly patients and frequently lead to treatment discontinuation.

Woo KA, Kim HJ, Yoo D, Choi JH, Shin J, Park S, Kim R, Jeon B. Patient-reported responses to medical treatment in primary dystonia. J Clin Neurosci. 2020 May;75:242-244. doi: 10.1016/j.jocn.2020.03.025. Epub 2020 Apr 2. PMID: 32249176.

Pirio Richardson S, Wegele AR, Skipper B, Deligtisch A, Jinnah HA, Dystonia Coalition I. Dystonia treatment: Patterns of medication use in an international cohort. Neurology. 2017;88(6):543-50.

5. Please elaborate on the prospective multicenter observational studies, specifying whether they were designed with two arms or three arms. Additionally, clarify when and where data on oral medication were included in the study.

Response:

In our previous multicenter observational study, we collected pre- and post-treatment data following BoNT-A administration, focusing on severity and utility scores for a cost-utility analysis. For comparison, we used the utility scores of new patients who had never received BoNT-A as the baseline "oral medication treatment" data. We added the data of oral medication treatment and source of data in table 1. Input parameters.

6. Please include the cost of oral medications in Table 1 to facilitate understanding, and ensure the source of this cost information is clearly indicated within the table.

Response: (Table1)

We have inserted the cost of oral medications in Table 1.

7. Please provide a citation for data on accidents resulting from severe blepharospasm

Response:

There are in introduction and table 1 input parameters.

Hwang WJ, Tsai CF. Motor vehicle accidents and injuries in patients with idiopathic blepharospasm. J Neurol Sci. 2014;339(1-2):217-9.

8. Please add the p-value to Table 3 for easier interpretation

Response:

In economic evaluation, the base-case result as shown in Table 3 is the incremental cost-effectiveness ratio (ICER), which is the ratio of difference in cost divided by the difference in effect between BoNT-A treatment and oral medication. To justify whether the new intervention is cost-effective, the estimated ICER would be compared with the ceiling ratio in each country. If the ICER is below the ceiling ratio, that new intervention is cost-effective. Therefore, p-value is not in need for the interpretation of the result.

9. Was the abbreviation "JRS" mentioned in full before it was used in the manuscript?

Response: (method, cohort population part)

Already added in method part, cohort population (first use).

10. Please cite for the definition of blepharospasm.

Response: (introduction part)

Already added in introduction part.

Defazio G, et al. Diagnostic criteria for blepharospasm: A multicenter international study. Parkinsonism Relat Disord. 2021 Oct;91:109-114. doi: 10.1016/j.parkreldis.2021.09.004. Epub 2021 Sep 8. PMID: 34583301; PMCID: PMC9048224

11. Please suggest ways to improve the sensitivity for utility of JRS in future analyses etc.

Response: (discussion part)

Might recruit more sample sizes OR collect long-term QoL data.

Reviewer #2: Thank you for your comments.

1. The authors should add more information in the introduction about the national burden of Blepharospasm in Thailand. This will justify the study question

Response: (Introduction part)

We revised introduction part to show the background of this study and research question clearer than the first draft, include national burden of blepharospasm in Thailand.

2. In the abstract, what does the author mean by hybrid model? This is not well-known in pharmacoeconomic studies. If they mean decision tree followed by Markov then please mention it.

Response: (abstract part)

We have revised the word from a hybrid model to a two-part model, which is similar to the method section.

A two-part model in this study represents the decision tree and Markov model.

3. Is there any population receive supportive care? It is important to mention what is the standard practice in Thailand.

Response: (Introduction part)

This is model-base study by use cohort population in the model. Use cohort population from multicenter study: supportive care (oral medical treatment) and BoNT-A treatment.

4. How you confirmed that the study sample is generalizable to Thailand population? Please add a table for patient demographics.

Response: (Introduction part)

The data were recruit from the 14 nationwide medical centers across the country. These centers represent the main providers of botulinum toxin treatments for blepharospasm in Thailand. The demographic data of reference source was showed in S1table.

We provide the table of demographic data of the patients in S1Table: Demographic data of patient references.

Reviewer #3: >> Thank you for your comments.

1. One of the issues I found was that the two treatment regimens for BoNT-A treatment were not separate nodes in the decision tree model but were instead combined. Since they are different alternatives of the BoNT-A treatment, each arm can be represented by their own node, with relevant probabilities calculated from the study data.

Response:

We have revised the Figure 1A to demonstrate the two treatment regimens for BoNT-A treatment.

2. Secondly, the decision tree for the comparator treatment assumed that there was no therapeutic effect of the comparator treatment. How is this different than assuming that the patient took no other treatment? Finally, the BoNT-A treatments were not adequately described which caused further confusion when the authors talked about months of treatment, and subsequent follow-up treatment.

Response:

We have changed the term “no BoNT-A treatment” to “oral medications” for the whole manuscript. We assume that patients who receive oral medications remain alive as the general population of the same age, but their quality of life might be worse than the general population (utility score = 0.75). For those who received BoNT-A treatment, outcome of treatment would be either response or non-response of BoNT-A treatment. Responded patients can be divided into a good-postpone group or a good group. The good-postpone group would receive BoNT-A treatment every 4 months or 3 times per year while the good group need BoNT-A every 3 months or 4 times per year. For the non-responded patients, they can be either a poor group or a poor-postpone/quit group. The poor group remains receiving BoNT-A treatment every 3 months or 4 times per year. However, the QoL would be worse than the responded group. The poor-postpone/quit group would receive BoNT-A treatment only once then switch to oral medications.

Since patients receive different frequency of BoNT-A treatment as above explanation, we decided to estimate monthly cost and outcome instead of cycle of BoNT-A treatment (every 3 months or 4 months).

3. Page 5, Line 108 to 109:

Please define terms before using abbreviations. What does JRS stand for?

Response:

Jankovic rating scale (JRS) was added in method part, cohort population (first use).

Secondly, the sentence needs to be rephrased for coherence. Do you mean, “Eligible patients had a total JRS score of 6 or higher, which indicate severe symptoms that affect daily living”.

Response:

Done.

4. Page 6, Line 112 to 114:

Could you please be more descriptive and specific about the interventions? The way this paragraph is written, it gives the impression that patients were given 30 units of onaBoNT-A per day for 12 weeks and up to a total of 16 weeks. It also gives the impression that patients were given 90 units of aboBoNT-A per day for 12 weeks and up to a total of 16 weeks.

Is that the dosage of this intervention? What is the route of administration? Is the treatment actually just one injection that is administered once, and a subsequent injection is administered after 12 or 16 weeks based on the patient response.

Response: (Method, Interventions and comparator part)

The study interventions were onabotulinumtoxinA (onaBoNT-A: Botox®) 30 units and abobotulinumtoxinA (aboBoNT-A: Dysport®) 90 units for a duration of approximately 12 to 16 weeks, depending on the patient’s response. Due to differences in formulation and potency, the units of onaBoNT-A and aboBoNT-A are not directly comparable, with an approximate conversion ratio of 1 unit of onaBoNT-A to 3 units of aboBoNT-A for blepharospasm treatment. BoNT-A is typically administered once every 12 to 16 weeks, with dosages adjusted based on individual patient responses. Each session involves intramuscular injections at 4 to 10 sites around both eyes, depending on condition severity and physician preference.

5. Page 6, line 115 to 119:

Which comparator treatments are considered in this study, and how are they incorporated into the model in the subsequent section of the manuscript?

Response:

The comparator in this study is oral medications. Patients who received oral medications remain alive as the general population at the same age group, but the QoL of patients would be worse than the general population who does not have the disease.

6. Figure 1A:

BoNT-A treatment arm: Why were both interventions (onaBoNT-A and aboBoNT-A) not included as separate arms in the decision tree mode? Is there no difference between the two?

Response:

We have revised Figure 1A to show both interventions. In our analysis, we assumed that onaBoNT-A and aboBoNT-A have similar effects based on findings from previous studies. Therefore, we did not include them as separate arms in the decision tree mod

---

## [Editor Report · Decision Letter 1]

11 Feb 2025

Cost-utility analysis of botulinum toxin type A versus oral drug treatment in patients with severe blepharospasm in Thailand

PONE-D-24-22961R1

Dear Dr. permsuwan,

We’re pleased to inform you that your manuscript has been judged scientifically suitable for publication and will be formally accepted for publication once it meets all outstanding technical requirements.

Kind regards,

Gang Qin, PhD, MD

Academic Editor

PLOS ONE

Additional Editor Comments (optional):

Thank you for addressing all the reviewers' comments. I am pleased to accept your manuscript for publication in Plos One. Congratulations!
---

## [Editor Report · Acceptance letter]

PONE-D-24-22961R1

PLOS ONE

Dear Dr. Permsuwan,

I'm pleased to inform you that your manuscript has been deemed suitable for publication in PLOS ONE. Congratulations! Your manuscript is now being handed over to our production team.

Kind regards,

on behalf of

Dr. Gang Qin

Academic Editor

PLOS ONE